# Situational Factors Impacting Harmful Behavior Towards Others Related to Mental Health in the Community and Their Associations: A Scoping Review Based on Systematic Reviews

**DOI:** 10.3390/healthcare13020152

**Published:** 2025-01-14

**Authors:** Issho Kobata, Yoshitomo Fukuura, Yuzaburo Kaba, Yukako Shigematsu

**Affiliations:** 1Department of Nursing, Kurume University Graduate School of Medicine, 777-1 Higashikushiharamachi, Kurume-Shi 830-0003, Fukuoka, Japan; 2Department of Nursing, School of Medicine, Kurume University, 777-1 Higashikushiharamachi, Kurume-Shi 830-0003, Fukuoka, Japan; fukuura_yoshitomo@kurume-u.ac.jp (Y.F.); kaba1208@med.kurume-u.ac.jp (Y.K.); shigematsu_yukako@med.kurume-u.ac.jp (Y.S.)

**Keywords:** harmful behavior toward others, violence, aggression, situational factors, scoping review

## Abstract

Background/Objectives: This study aimed to identify factors associated with harmful behavior toward others based on existing research. Methods: This scoping review focused on individuals at risk of harming others due to mental health issues, with the target population encompassing three settings: the community, inpatient facilities with frequent admissions and discharges, and healthcare settings where medical treatment is sought. A scoping review was conducted according to the Preferred Reporting Items for Systematic reviews and Meta-Analyses extension for Scoping Reviews. The terms violence, aggression, problem behavior, and workplace violence were used to search for related literature, subsequently selecting systematic reviews. Results: A total of 24 papers were ultimately included. From the included papers, background factors (demographic, personal history, and clinical aspects); situational factors (social connection status, daily life status); psychological factors; antecedents of harmful behavior; and triggers of harmful behavior were extracted as factors associated with harmful behavior. Conclusions: Our results indicate that background and situational factors lead to harmful behavior toward others, disruptions in the harmony between these factors cause disturbances in psychological processes, and harmful behavior toward others is triggered by stimuli that promote such behavior. Considering that all studies reviewed herein involved inpatients in medical settings, further research is required to identify the factors associated with harmful behaviors occurring in the community.

## 1. Introduction

Harmful behavior toward others, such as committing actual physical violence against another person or property or making specific and imminent verbal threats, can cause emotional or physical trauma [1,2]. However, no generalized standards for harmful behavior toward others yet exist, and defining such behaviors for the purpose of establishing laws has proven difficult [3].

In Japan, the Mental Health and Welfare Law defines “other harmful acts” as “acts that harm the life, body, chastity, honor, property, or social interests of others, such as murder, injury, assault, sexual problem behavior, insult, damage to property, robbery, extortion, theft, fraud, arson, and tampering with fire”. For persons at risk of causing other people harm due to mental disorders, compulsory hospitalization is enforced by order of the prefectural governor when necessary to ensure the safety of the person concerned, his/her family, and local residents and to provide appropriate treatment [4,5]. Mental health welfare professionals in the community are charged with managing people at risk of causing harm to others, including emergency response and crisis intervention for those at risk of causing harm to others. Mental health welfare professionals working at public health centers, which are the frontline public health organizations, facilitate involuntary hospitalization, if necessary, after examination by a designated mental health physician based on a police report, etc. [6]. However, if the police report indicates no risk of harm to others even after examination by a designated mental health doctor or if the patient was violent but had calmed down by the time the police arrived, the patient would not be subjected to involuntary hospitalization but would instead be left with the mental health welfare professionals who would be managing the case thereafter. Mental health welfare specialists stationed in various municipalities provide consultation support for mental health issues and respond to consultations from family members or neighbors affected by individuals who shout, throw things, cause disturbances, or act violently [6]. Numerous individuals with mental disturbances currently receive treatment or therapy. Unfortunately, several people with mental disorders commit acts of violence during treatment or after treatment interruption, which cannot be controlled by family members or people surrounding them, despite noticing the worsening mental symptoms and changes in behavior prior to committing acts of violence. Hence, responding to acts of violence in hospitals and local communities is becoming increasingly important [7,8].

Several countries worldwide, including many European countries, the United States, Oceania countries, and Asian countries, among others, have taken measures, such as involuntary hospitalization, to control those at risk of harming others based on their psychiatric symptoms [5], given the rising need for managing such cases within the community. In addition, aggression and violence toward others have become a major problem in the medical field, such as psychiatric wards. Considering that acts of harm in medical settings not only cause injury to staff and other patients but also affect staff morale and the effectiveness of treatment provided to other patients [9], research has sought to clarify the characteristics of persons at high risk of committing acts of harm [9,10,11,12,13,14,15]. Accordingly, studies have found that risk factors for aggression and violence in medical settings can be categorized into internal and external factors. Internal factors include clinical factors (e.g., schizophrenia and bipolar disorder); personal factors (e.g., impulsivity, hostility, and poor insight); historical factors (e.g., past violence and low social class); and drug and alcohol abuse. Meanwhile, external factors include environmental factors (e.g., the ward environment), hospitalization factors (e.g., involuntary hospitalization and length of stay), and relational factors (e.g., interpersonal relationships with staff) [10]. Six domains identify the key influences over conflict and containment rates within psychiatric wards: the patient community, patient characteristics, the regulatory framework, the staff team, the physical environment, and outside hospital. Furthermore, triggers of conflict, such as violence, have been identified, including refusal of patient requests, arguments with friends and family, and curtailment of freedoms [7].

Both health facilities and communities need to establish appropriate responses to persons who may cause harm. In the community, the internal factors of persons who may cause harm often cannot be determined when providing a response, and their living environment and relationships with other people vary. Previous studies on harmful behavior toward others have focused on cases occurring in medical facilities, specific diseases, and the acute stage; thus, factors associated with harmful behavior toward others in the community have yet to be fully clarified. We believe that clarifying the factors associated with harmful behavior toward others will help mental health welfare professionals understand such factors and assess situations that may cause harmful behavior toward others, including the degree of risk. To determine the appropriate response to persons who may cause harm in the community, clarifying factors associated with harmful behavior based on existing research is necessary. Therefore, we conducted a scoping review to obtain relevant knowledge and factors related to harmful behavior that could help in the investigation of the actual situation of harmful behavior in the community.

## 2. Materials and Methods

### 2.1. Study Design

Our research methodology involved a scoping review that aimed to outline the key concepts and available evidence underlying the relevant factors leading to harmful behavior toward others [16]. This study was planned and conducted in accordance with the PRISMA (Preferred Reporting Items for Systematic reviews and Meta-Analyses extension for Scoping Reviews) guidelines [17]. Before the start of the search, a review protocol was entered into the framework database (https://osf.io/g28t3) (accessed on 26 November 2024).

Our research question was as follows: “What are the factors that cause people to commit acts of harm?” Patient: A person who commits a harmful act in the community (a person who is at risk of committing a harmful act). Concept: To identify factors associated with harmful acts. Context: Exhaustive factors associated with committing a harmful act, according to the research question.

This scoping review focused on individuals living in the community who are at risk of harming others due to mental health issues. Therefore, I have defined the target population to include three settings: the community where these individuals reside, inpatient facilities due to frequent admissions and discharges, and healthcare settings where they seek medical treatment.

### 2.2. Literature Search and Identification

In this study, violence, aggression, and problem behavior were indicated as harmful behavior toward others. Our literature search was conducted using the PubMed and Scopus databases.

The formula “Violence OR Aggression OR Problem Behavior OR Workplace violence And systematic review” was used to search PubMed, whereas the formula “Problem Behavior OR Workplace violence And systematic review” was used to search Scopus. Only studies published in English were searched, with no restrictions on the year of publication. Although a considerable amount of data is available on harmful behavior in medical facilities and among patients with specific diseases, obtaining comprehensive data is necessary to examine factors associated with harmful behavior in the community. Therefore, a systematic review was conducted to accumulate comprehensive data from various sources.

A similar approach was used in the identification of all papers included in this study by setting the eligibility criteria before study initiation. Moreover, two reviewers independently determined whether or not the literature was acceptable [16] based on “patient” (i.e., the person who commits the harm), “concept” (i.e., a description of relevant factors leading up to the harm), and “context” (i.e., an exhaustive description of the harmful behavior). For the first screening, two researchers independently evaluated the titles and abstracts to determine whether they satisfied the eligibility criteria. Cases of disagreements among the researchers were resolved through consensus after discussing the results among themselves. In the second screening, two researchers independently read the full texts of the identified literature to confirm whether the papers satisfied the eligibility criteria. Similarly, differences in opinion were resolved through consensus after discussions among the researchers; after which, a final decision on whether to include the paper was made.

### 2.3. Data Extraction and Integration of the Results

The title, author, year of publication, purpose, subject, scope, and main results and discussions were extracted and organized. The main research results extracted were divided into the following categories: literature related to general behavior, such as violence and aggression, literature related to behaviors in patients with psychiatric disorders, and literature related to rating scales of factors associated with harmful behaviors toward others.

The following procedure was used to qualitatively integrate descriptions of factors related to harmful behavior in the literature [16]. All texts and rating scales obtained from the search results were carefully read to extract descriptions of the factors related to harmful behavior toward others. The extracted research results were coded by expressing them in a single sentence while ensuring that the meaning of each group was not compromised. Codes, subcategories, and categories were generated by grouping them based on their differences and commonalities. To ensure the validity of the analysis, all authors evaluated the consistency of the generated codes, subcategories, and categories in the aforementioned process and obtained unanimous agreement. The entire set of generated categories was reviewed; after which, the contents of the extracted categories were examined. We then discussed the relationships among the categories and between categories and harmful behavior toward others.

## 3. Results

### 3.1. Overview of the Included Studies

The literature search identified 7045 papers. The literature for this scoping review consists of articles that examine relevant factors contributing to harmful behavior toward others in the community, particularly in situations where mental health welfare professionals are likely to be involved. Literature on child and elder abuse, literature on drug treatment and seclusion restraints, literature on staff-to-patient and staff-to-staff violence, and literature not related to community or mental illness were excluded, because they did not fit the theme and conceptual scope of this review. After excluding 6879 and 142 papers in the primary and secondary screening, respectively, 24 papers were ultimately included (Figure 1).

Among the papers included herein, all of which were published between 2007 and 2023, 6 focused on general behaviors such as violence and aggression [3,8,12,13,18,19], 13 articles focused on behaviors among patients with psychiatric disorders [2,9,11,15,20,21,22,23,24,25,26,27,28], and 5 focused on rating scales for factors related to harmful behaviors toward others [1,29,30,31,32]. The scope of our study included papers involving inpatients, those focusing on the community, and those in medical settings.

Table 1 provides an overview of the included papers organized according to title, author, year of publication, purpose of the study, subject matter, scope, and summary of the results and discussion.

### 3.2. Factors Associated with Harmful Behavior Toward Others

From the target literature (Table 1), 22 subcategories and 7 categories were extracted (Table 2), using the relevant factors for harmful behavior toward others as codes.

{Background factors (demographic and personal history)} were extracted from the following seven subcategories: <Demographic and environmental factors>, which includes four codes, such as [age]; <history of self-injurious/other harmful behavior>, which includes two codes, such as [history of violence]; <apprehension/criminal history>, which includes three codes, such as [history of criminal/criminal activity]; <violence victimization/experienced abuse>, which includes three codes, such as [experience of violence victimization]; <family history (mental illness/substance use problems/criminal involvement)>, which includes three codes, such as [family history of mental illness]; <intelligence/history of school>, which includes two codes, such as [intelligence]; and <secure attachment and conduct problems during childhood>, which includes three codes, such as [secure attachment in childhood/stable nurturing environment].

{Background factors (clinical aspects)} were extracted from the following four subcategories: <mental illness/organic disorder (schizophrenia/personality disorder/mood disorder)>, which includes 10 codes, such as [history of mental illness]; <substance use problems (alcohol/drugs)>, which includes 4 codes, such as [previous and/or current drug abuse]; <treatment status (treatment motivation/compliance/sensitivity)>, which includes 6 codes, such as [compliance/treatment motivation]; and <positive symptoms (e.g., hallucinations and delusions)>, which includes 2 codes, such as [hallucinations/delusions].

{Situational factors (daily life status)} were extracted from the following two subcategories: <work/economic situation>, which includes three codes, such as [work/work training], and <basic self-care independence>, which includes three codes, such as [self-care/ability to perform daily chores].

{Situational factors (social connection status)} were extracted from the following two subcategories: <presence and interaction with family and friends>, which includes three codes, such as [cohabitee], and <social connections and access to public and private support systems>, which includes eight codes, such as [social support/network].

{Psychological factors (thinking, cognition, and personality)} were extracted from the following five subcategories: <self-awareness/foresight>, which includes four codes, such as [understanding of mental illness and treatment needs/insight into illness and/or behavior]; <ability to put yourself in other people’s shoes (cooperation/empathy)>, which includes nine codes, such as [empathy/cooperativeness]; <impulsivity/emotional instability>, which includes four codes, such as [impulsivity/poor behavioral controls]; <vulnerability to stress/coping skills>, which includes two codes, such as [leisure/recreational activities]; and <abnormalities or distortions in personality, thinking, and behavior>, which includes five codes, such as [cognitive distortion/distortion of thought].

{Antecedents of harmful behavior} were extracted from one subcategory, namely, <imminent risk of harm to others and threatening behavior/attitude>, which includes nine codes, such as [verbal threats and physical threats].

{Triggers of harmful behavior} were extracted from one subcategory, namely, <triggers of the harmful behavior>, which includes five codes, such as [stressing circumstances].

## 4. Discussion

The current scoping review identified seven categories that could be considered factors associated with harmful behavior toward others (Figure 2).

Initially, the intricate intertwining of background and situational factors form the framework for the harmful behavior. Accordingly, a disturbance in the harmony between background and situational factors shifts the psychological state of an individual to one in which thoughts, cognitions, and personalities (i.e., psychological factors for harmful behavior) emerge. Further deterioration of the psychological state in situations where psychological factors are emerging brings out behaviors and attitudes antecedent to harmful behavior toward others. Consequently, the emergence of behaviors and attitudes that serve as precursors to harmful behavior can lead to actual harmful behavior toward others in the presence of some triggers.

### 4.1. Background of Harmful Behavior Toward Others

Background and situational factors do not directly contribute to harmful behavior toward others.

The background factors {demographic and personal history} are cultivated in an individual throughout their life and serve as the basis for related factors that influence other related factors. Clinical aspects, which are factors related to mental illness and treatment status, are the most important factors leading to harmful behavior toward others [23]. Factors related to personal history, such as family history, intelligence, and childhood upbringing, influence the incidence and symptoms of mental illness [33,34,35]. In addition, clinical aspects, such as symptoms of mental illness and lack of treatment compliance, have been associated with difficulties in establishing relationships and social ties [36,37,38], employment difficulties, deprivation, and the inability to build an independent life [38,39]. Thus, background factors form {clinical aspects} based on {demographic and personal history} and influence situational factors.

The situational factor {daily living status} determines the stability of a person’s daily life, including financial problems and self-care situation, whereas {social connection status} determines the presence or absence of family members and surrounding support persons [37]. Interventions targeting both of these factors have been shown to reduce the risk of committing harmful behavior [15]. Given that the quality of life of persons with mental illness is related to their social networks [40], situational factors are interrelated with their inability to build relationships with those around them due to the lack of a stable daily life and the inability to lead an independent daily life resulting from the lack of necessary support caused by weak social ties. Furthermore, social isolation creates a situation that facilities failure of symptom control and worsening of mental illness [38]. In this way, the factors such as {daily life status} and {social connection status} become interrelated, creating situational factors that contribute to harmful behavior and influence background factors such as clinical aspects.

Should background and situational factors achieve harmony, such as in individuals capable of self-care with a surrounding support system, it is possible to remain in the “calm period”, where one can continue with daily life without the appearance of psychological factors for harmful behavior even with worsening mental symptoms. However, when situational factors worsen due to clinical deterioration, such as a worsening of psychiatric symptoms, the patient enters the “risk period” characterized by a disturbance in the harmony between background and situational factors and the emergence of psychological factors that lead to harmful behavior toward others.

### 4.2. Disorders in the Psychological Processes: Psychological Factors, Antecedents of Harmful Behavior

Disorders in the psychological process is a psychological state (psychological factors and antecedents of harmful behavior) that emerges when background and situational factors for harmful behavior become discordant.

{Psychological factors} refer to a state in which the harmony between background and situational factors for harmful behavior is disrupted, creating distortions in thoughts, cognitions, and personalities. Psychological states, such as a lack of insight and cooperation and impulsivity, are caused by clinical aspects such as mental disorders and are formed in relation to social and daily functioning [40,41]. In this situation, thoughts, cognitions, and personalities (i.e., psychological factors for harmful behavior) emerge due to the deterioration of clinical aspects, such as psychiatric symptoms, which are exacerbated by situational factors, such as a lack of social support and the inability to lead an independent life, and a disruption in harmony between background and situational factors for harmful behavior.

{Antecedents of harmful behavior}, which refer to behaviors and attitudes that are precursors to harm, emerge when psychological factors, such as thoughts, cognition, and personality, are present and are further aggravated by the psychological state. The presence of hyperactivity, agitation, and accelerated motor activity in persons with mental illness has been associated with imminent aggression, and antecedents to harmful behavior toward others may be the only visible sign observed in patients at imminent risk [42]. Behaviors and attitudes that are antecedents to harmful behavior toward others are identified as dynamic factors in rating scales and can be targeted by intervention to reduce the risk of harmful behavior [15]. In this situation, a person who is impulsive or lacks insight exhibits behaviors such as throwing things or irritable attitudes that result from worsening psychological conditions due to stress.

During the “risk period” when disturbances in the psychological process occur, further deterioration in the psychological state leads to the “precursor period”. In addition, exposure to triggers of harmful behavior toward others during the “precursor period” leads to the “emergence period” when the harmful behavior emerges.

### 4.3. Triggers of Harmful Behavior (Triggers)

{Triggers of harmful behavior} refer to events or objects (triggers) that serve as triggers for the transition from a situation in which psychological factors (i.e., thoughts, cognitions, and personalities) emerge to that in which behaviors and attitudes that serve as precursors to harmful behavior emerge or from a situation in which behaviors and attitudes that serve as precursors to harmful behavior are observed to that in which harmful behavior toward others occurs. Considering that external stressors increase stress and excitement and induce the emergence of aggression [14], external factors, such as situations and events, can serve as triggers for the occurrence of harmful behavior toward others. Triggers of harmful behavior include factors that promote such harmful behavior by affecting disturbances in psychological processes.

Exposure to triggers of harmful behavior, such as a loss-like life event or encounter with a certain person in the presence of behaviors and attitudes that act as precursors to harmful behavior, such as being irritable or verbally threatening, can lead to actual harmful behavior, such as physical violence or problematic behavior.

### 4.4. Limitations

This scoping review was conducted using two databases for the search. As the review focused on systematic reviews, primary research articles and special feature articles were excluded. Additionally, a large proportion of the included literature originated from European and American countries. Therefore, there is a potential for selection bias.

The current study identified factors associated with harmful behavior toward others through a systematic review of studies on harmful behavior and discussed the relevance of these factors. However, the literature included in this review is based on inpatients in medical settings. Hence, further investigations are needed to clarify the associated factors for harmful behavior occurring in the community. We believe it necessary to focus on situational factors related to daily life and social connections, antecedents of harmful behavior, and triggers of harmful behavior in the community, which may differ from those in medical settings.

Future studies should investigate the actual conditions for harmful behavior in the community, as well as support for individuals exhibiting such behavior, based on the related factors and their associations presented herein while refining the model for factors related to harmful behavior to make it suitable for use in community settings.

## 5. Conclusions

Seven categories of factors related to the harmful behavior were extracted from the literature and rating scales: background factors (demographic and personal history and clinical aspects), situational factors (social connection status and daily life status), psychological factors, antecedents of the harmful behavior, and triggers of the harmful behavior.

Our results suggest that background and situational factors form the background factors that lead to harmful behavior, that a disturbance in the harmony between these factors causes a disturbance in the psychological process, and that harmful behavior toward others is triggered by stimuli that leads to such behavior.

Considering that all the studies reviewed herein involved inpatients in medical settings, further research is required to identify the factors associated with harmful behaviors occurring in the community.

## Figures and Tables

**Figure 1 healthcare-13-00152-f001:**
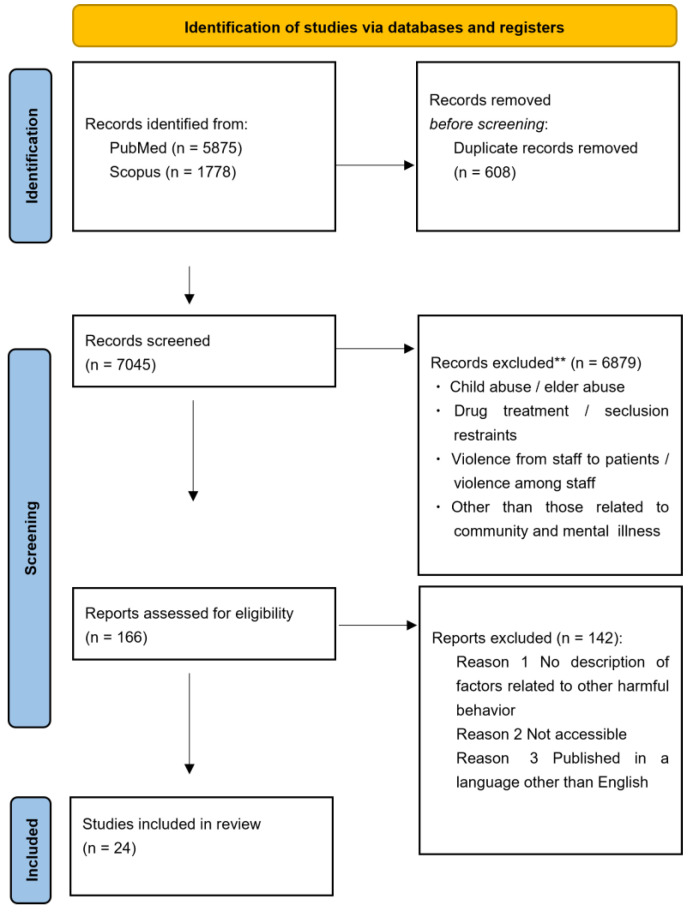
PRISMA flowchart for the article selection process. ** Based on title/abstract screening.

**Figure 2 healthcare-13-00152-f002:**
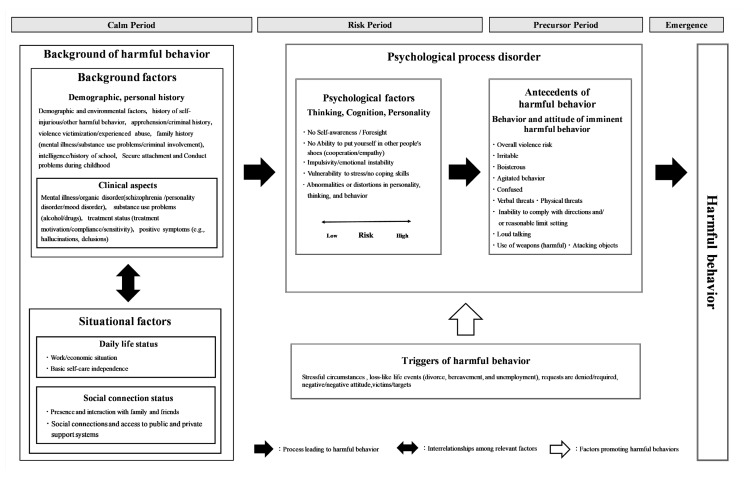
Relevance model of the factors associated with harmful behavior toward others.

**Table 1 healthcare-13-00152-t001:** Summary of the included literature.

〇 Summary of Studies on Harmful Behavior Toward Others in General (e.g., Aggression, Violence)						
No.	Title	Author (Year)	Aim	Subjects	Scope *	Summary of Results and Discussion
1	A review of effective interventions for reducing aggression and violence	James McGuire (2008) [3]	To research on the outcomes of interventions designed to reduce personal violence and summarize what has emerged from that work while focusing on identifying the most effective approaches to the problem that have been discovered to date.	Identify articles on the nature of violence, issues in violence intervention research, reducing criminal recidivism, violence meta-analysis, interventions to reduce violence, and conceptualizing violent behavior.	C	A literature review of the nature of violence was conducted from the perspectives of definitional problems, spatial and temporal variations, individual continuity, and expressive and instrumental aggression. The definition of violence is an unstable and unsatisfactory process and varies widely depending on the geographical situation and the social conditions of the time. Some individuals continue to engage in violence. Two types of aggression were identified: expressive and instrumental. The review focused on recidivism prevention, interventions for young offenders, and interventions to reduce violence and aggression, with suggestions that some interventions could reduce the occurrence of patterns of violence and aggression. Emotional self-management, enhancement of interpersonal skills, and social problem-solving training were found to be effective with fairly high reliability.
2	Situational variables related to aggression in institutional settings	Elena Welsh, Shannon Bader, Sean E. Evans. (2013) [19]	To outline the salient areas discussed in situational factor research on aggression and to present a recent theoretical model that integrates these factors.	Articles published in PsychInfo and MEDLINE databases were searched. Various key words, such as inpatient, hospital, psychiatric, and prison, were used to identify research conducted in the settings of interest, with the words aggression and violence narrowing the search. Only studies published in English were used, and no attempt to search for unpublished research was made. specific risk assessment measures administered to patients, or patient factors related to violence were not reviewed.	I	Regarding temporal factors, the highest rates of violent incidents occurred in the morning, whereas the most serious incidents in the facility occurred in the afternoon. The specific effect ward crowding had on aggression remained unknown. Characteristics that predisposed patients to violence were as follows: haphazard and unreliable schedules and routines, poor teamwork and distribution of responsibility, limited psychiatrist availability, patients perceived as dangerous and to be feared, little interaction between patients and staff, and few therapeutic activities provided. The following are some of the most common reasons for the lack of therapeutic activities.
3	The assessment and management of the violent patient in critical hospital settings	Carl L. Tishler, Natalie S. Reiss, John Dundas. (2013) [13]	To provide an up-to-date review of the literature regarding the management of violence in the emergency department for trainees and experienced clinicians.	This narrative review was largely derived from research articles and reviews published since 2000. A systematic search of electronic databases was conducted for review articles or studies examining patient violence and aggression in critical care settings.	I	The paper sumarizes research on risk factors for violent and aggressive behavior. For prevention and intervention, aggressive use of medications is justified in high-risk situations, with restraints to be use as a last resort.
4	Incidence and factors associated with substance abuse and patient-related violence in the emergency department: A literature review	Sabine Kleissl-Muira, Anita Raymond, Muhammad Aziz Rahman. (2018) [12]	To investigate the incidence of workplace violence in the emergency department and its association with substance abuse, as well as to identify factors associated with workplace violence, such as perpetrator characteristics and environmental factors.	A literature search using the search terms such as drug abuse, alcohol abuse, regional, rural, emergency department, violence, and aggression was conducted using the following databases: CINAHL, Cochrane Library, PsychINFO, Ovid Medline, PubMed, and Google Scholar. Literature published in English reporting violent or aggressive emergency department patients was included. Excluded articles that did not address emergency department violence related to substance abuse or literature on pediatric populations. Fourteen studies were included.	I	Incidents of violence ranged from 2.8 to 10.3 per 1000 emergency patients, with up to 66% related to drug or alcohol exposure.
5	Neighborhood Interventions to Reduce Violence	Michelle C Kondo, Elena Andreyeva, Eugenia C South, John M MacDonald, Charles C Branas. (2018) [8]	To examine studies that use quasi-experimental or experimental designs to compare violence outcomes for treatment and control groups before and after a change is implemented in the built environment.	Articles on interventions and findings of quasi-experimental studies on neighborhood environments and violence outcomes were searched. Studies on violent crimes were included but not those that focused exclusively on property crimes, drug crimes, or nuisance crimes.	C	Theories linking neighborhoods and violence include the Routine Activities Theory and the Broken Window Theory. Neighborhood environmental interventions include housing improvements, land use and zoning changes, depletion restoration, greening of vacant lots, transportation improvements, and school improvements. Neighborhood interventions required strong partnerships among diverse stakeholders, including municipalities, nonprofit organizations, community groups, academic institutions, and funders.
6	Risk factors for interpersonal violence: an umbrella review of meta-analyses	Seena Fazel, E. Naomi Smith, Zheng Chang and John Richard Geddes. (2018) [18]	To examine the strength and population effect of modifiable risk factors for interpersonal violence, as well as the quality and reproducibility of the research evidence.	Three databases, namely PsycINFO, Medline, and Global Health, were searched followed by targeted searches in Google Scholar and PubMed. The literature search was conducted using the keyword violence in combination with risk factors and publication search terms. Systematic reviews and meta-analyses on a wide range of violence outcomes, including assault, violent crime, and sexual violence, were included; studies on verbal aggression, minor criminality, and antisocial behavior were excluded. A total of 22 studies were included.	C	Many important risk factors for violence are modifiable, and violence prevention strategies should incorporate identification, assessment, and treatment of mental disorders.
* Letters in the scope items indicate the following; I: inpatient; C: community; M: medical field (no distinction between outpatient and inpatient).						
**〇 Summary of Studies on Harmful Behavior Toward Others Among Patients with Mental Illness**						
**No.**	**Title**	**Author (year)**	**Aim**	**Subjects**	**Scope** *****	**Summary of results and discussion**
7	Violence and aggression: a literature review	P Woods, C Ashley. (2007) [15]	To provide a background for a small pilot study that introduced violence measurement instruments to assist in the development of nursing practice in an acute psychiatric unit.	Multiple databases were searched, focusing on publications since 1994: CINAHL, Ovid Healthstar, Ovid MEDLINE (R), EMBASE, and PsycINFO. The search used the following four groups of key word alternatives (used in truncated form to allow for ending permutations) in combination with each other: violence, aggression, dangerous; prediction, assessment, factor, risk, issue, cause, reason; mental, psychiatric; inpatient, short-term, acute, admission.	I	Violence by psychiatric inpatients has important consequences for the treatment environment, other patients, and staff. Methods for predicting of psychiatric violence have been lacking, and approaches to risk assessment need to include both actuarial (historical factors) and individual risk factors (history, severity, and pattern of occurrence of behavior) associated with the behavior. Given that risk assessment of aggression cannot be based on clinical diagnosis alone, factors related to violence in psychiatric inpatients were categorized into four domains: dispositional, contextual, clinical, and historical. Some argue that gender and age should not be included as risk factors. Risk factors for violence can be categorized as static (unchangeable/changeable) and dynamic (changeable).
8	Neurological disorders and violence: a systematic review and meta-analysis with a focus on epilepsy and traumatic brain injury	Seena Fazel, Johanna Philipson, Lisa Gardiner, Rowena Merritt & Martin Grann. (2009) [23]	To systematically review and meta-analyze the research literature on the association of common neurological disorders and violence. To investigate whether an association of specific neurological illnesses with violence compared with the general population and what risk factors are associated with violence in these disorders.	Studies of the association of neurological illnesses and violence reported between January 1966 and August 2008 were sought by searches of electronic bibliographic databases (MEDLINE, EMBASE, PsycINFO, CINAHL) using combinations of keywords relating to the neurological illnesses (headache/migraine, stroke, CNS infection, Parkinson’s disease, essential tremor, epilepsy, head injury, multiple sclerosis, chronic fatigue syndrome, brain tumours, Huntington’s disease) and to violence. excluding those with anger and impulsivity as outcomes and dementia.20 studies were included.	M	Evidence was limited, and methodological quality varied. The risk of violence appeared to differ between patients with epilepsy and traumatic brain injury and unaffected controls. Epilepsy was inversely associated with violence, whereas brain injury was associated with a slightly increased risk.
9	Schizophrenia and violence: systematic review and meta-analysis	Seena Fazel, Gautam Gulati, Louise Linsell, John R Geddes, Martin Grann. (2009) [22]	To conduct a systematic review and meta-analysis with the aim of resolving the variability (heterogeneity) of conclusions reached in studies on the association between violence and schizophrenia and other psychoses and between substance abuse and violence.	Computerized searches of Medline, Embase, and Psycinfo were performed for studies published from January 1970 to February 2009 using the terms viol*, crim*, homicide, schiz*, severe mental illness, major mental disorder, psychos*, and psychot*. Studies that did not perform any comparison with the general population or with other psychiatric diagnostic groups were excluded. A total of 20 studies were included.	M	The results of studies comparing violence in patients with schizophrenia or other psychosis to those without mental disorders have been considerably heterogeneous, but evidence sugggest an increased risk of violent outcomes (violence, violent crime, and homicide) in patients with schizophrenia or other psychosis. Comorbid conditions with substance use disorders substantially increase this risk.
10	Aggression in psychiatry wards: a systematic review	Cesare Maria Cornaggia, Massimiliano Beghi, Fabrizio Pavone, Francesco Barale. (2011) [20]	To identify individuals more likely to become violent during hospitalization.	PubMed, Embase, and PsychInfo databases were searched for papers written in English, Italian, French, or German published between 1 January 1990 and 31 March 2010 using the key words “aggress*” (aggression or aggressive) “violen*” (violence or violent) and “in-patient” or “psychiatric wards”. Exclusion criteria include a specific psychiatric diagnosis other than psychosis, adolescents or the elderly, men/women only, personality disorders and mental retardation, treatment strategies, and studies conducted in non-acute settings. Overall, 109 studies were included.	I	Episodes of violence on the ward were more common among young patients with schizophrenia, those with neurological problems, and those in a chaotic environment. The results suggest that good communication, harmony among staff, and the presence of male nurses are effective in preventing aggression.
11	A review and meta-analysis of the patientfactors associated with psychiatric in-patient aggression	C. Dack, J. Ross, C. Papadopoulos, D. Stewart, L. Bowers. (2013) [11]	To combine the results of earlier comparison studies on aggressive and non-aggressive psychiatric inpatients to quantitatively assess the strength of the association between patient factors and aggressive behaviour. To identify differences between patients who were repetitively aggressive and those who were only aggressive once during their admission, as well as explore differences according to setting (acute vs. forensic wards).	A search was conducted using the following databases: MEDLINE, PsychInfo, Cochrane Clinical Trials, EMBASE Psychiatry, CINAHL and DARE, and the following keywords: (psychiat* or mental*) and (hospital or ward or inpatient or in-patient) and (aggressi* or violen*). No attempt was made to search for unpublished results. Peer-reviewed articles published in English that examined aggression in hospitalized patients were included. Articles containing secondary data, non-empirical data, or pediatric inpatient data were excluded. Overall, 44 studies were included.	I	Female inpatients with a history of substance abuse or violence were more likely to become aggressive repeatedly than to become aggressive once during hospitalization. Staff need to consider dynamic factors, such as the patient’s current condition and situation, to reduce aggression in hospitalized patients.
12	Angry affect and violence in the context of a psychotic illness: a systematic review and meta-analysis of the literature	Shuja Reagu, Roland Jones, Veena Kumari, Pamela J Taylor. (2013) [26]	To conduct a systematic review of all published studies on anger and violence in the context of schizophrenia or similar psychoses.	A search of online databases Medline, Embase, Psycinfo, DARE and the CENTRAL from inception till January 2012 was performed using the following search terms: delusion*, psychos*s, psychotic*, schizophreni* for schizophrenia and related psychoses; anger, angry, and rage for anger; and aggressi*, violen*, assault, and homicide for violence. A total of 11 studies were included.	M	Patients with schizophrenia who acted violently were reported to have increased anger scores. Significant associations were observed between anger and violent behavior in the context of psychotic disorders.
13	Risk Factors for Violence in Psychosis: Systematic Review and Meta-Regression Analysis of 110 Studies	Katrina Witt, Richard van Dorn, Seena Fazel. (2013) [28]	To determine the direction and strength of the association between risk and protective factors for violent outcomes in patients with psychosis.	We conducted a systematic review and meta-analysis of studies published in 6 electronic databases (CINAHL, EBSCO, EMBASE, Global Health, PsycINFO, PUBMED) and Google Scholar using keywords that were inclusive for psychosis (e.g., schiz*, psych*, mental*) and violence (e.g., viol*, aggress*, crim*, offend*, danger*, hosti*). We included studies that reported factors associated with violence in adults diagnosed with schizophrenia or other psychoses and excluded those that examined only risk factors for childhood violence or those whose samples did not distinguish between violent offenders and non-offenders.A total of 110 studies on 8439 violent patients were identified, among whom 45,533 had a mental illness.	M	More than 80% of the patients were diagnosed with schizophrenia, followed by bipolar disorder. Static factors were also examined, but the strongest factor was criminal history. The association remained consistent in studies that limited the results to serious violence, and the direction of the association remained the same for both inpatients and community patients.
14	Prevalence and Risk Factors of Violence by Psychiatric Acute Inpatients: A Systematic Review and Meta-Analysis	Laura Iozzino, Clarissa Ferrari, Matthew Large, Olav Nielssen, Giovanni de Girolamo. (2015) [9]	To conduct a systematic meta-analysis to estimate the pooled rate of violence, in terms of period prevalence, in acute psychiatric wards and explore the aggregate-level ward characteristics that might explain the variation in the reported rates of violence between wards.	Studies were identified by searching the electronic databases PubMed, Scopus and Cumulative Index to Nursing and Allied Health Literature (CINAHL). Included studies that reported on the proportion of adult patients admitted to acute psychiatric wards in high-income countries who committed at least one act of violence during their stay. Articles in which the number of patients who committed physical violence could not be ascertained, studies conducted in forensic hospitals or wards, studies conducted in residential facilities other than hospitals, and studies conducted on adolescent and geriatric patients were excluded. A total of 35 studies with 23,972 inpatients were included.	I	Around one in five patients admitted to an acute psychiatric unit in a high-income country was found to have committed an act of physical violence during their stay. Drug use may be mediated by psychiatric symptoms and social factors.
15	The relationship between paranoia and aggression in psychosis: A systematic review	Hannah Darrell-Berry, Katherine Berry, Sandra Bucci. (2016) [21]	To examine the relationship between paranoia and aggression in the context of schizophrenia, taking into account study quality, in order to enhance our understanding of the mechanisms through which some people diagnosed with schizophrenia are aggressive and violent.	An electronic database search of Ovid MEDLINE, PsycINFO, Embase, CINAHL, PubMed, and Web of Science was conducted (from inception to November 2014) using keywords related to schizophrenia, paranoia, and aggression. Studies using paranoia/psychiatric scales were included, whereas those using aggression scales were excluded. Overall, 15 studies were included.	IC	Mixed support for the association between paranoia and aggression in both inpatient and community settings. Aggression was linked to delusions and self-threatening delusions. Delusional individuals may use aggression as a safety behavior to maintain their own safety or prevent threats. Aggressive responses to perceived threats may be driven by a desire for retaliation.
16	How effective are risk assessments/measures for predicting future aggressive behaviour in adults with intellectual disabilities (ID): A systematic review and meta-analysis	Rachael Lofthouse, Laura Golding, Vasiliki Totsika, Richard Hastings, William Lindsay. (2017) [25]	To systematically review evidence on the efficacy of assessments for managing the risk of physical aggression in adults with intellectual disabilities (ID).	A literature search was conducted for studies published in the databases PsycINFO, EMBASE, MEDLINE, Web of Science, and Google Scholar using a combination of keywords: intellectual disability (intellectual disability, learning disability, developmental disability, mental retardation) and risk assessment. Search was restricted to studies with aggression as the outcome variable.In total, 14 studies were included.	M	The majority of the studies used a scale to assess the risk of aggression. The risk rating scale can significantly predict aggression in the future.
17	Preventing and De-escalating Aggressive Behavior Among Adult Psychiatric Patients: A Systematic Review of the Evidence	Bradley N Gaynes, Carrie L Brown, Linda J Lux, Kimberly A Brownley, Richard A Van Dorn, et al. (2017) [2]	To compare the effectiveness of strategies for preventing and de-escalating aggressive behaviors among psychiatric patients in acute care settings, including interventions for reducing the use of seclusion and restraint.	We searched MEDLINE (via PubMed), Embase, the Cochrane Library, Academic Search Premier, PsycINFO, and CINAHL (Cumulative Index to Nursing and Allied Health Literature). The literature search focused on comparative studies on de-escalation strategies (seclusion, restraint, or alternatives to seclusion or restraint) for adult patients with psychiatric disorders or severe psychiatric symptomatology who are at risk of, or present with, aggressive behavior across various acute care settings. Studies restricted to patients with dementia were excluded. A total of 17 studies were included.	I	Staff training, multimodal programs, risk assessment interventions, environmental and group psychotherapy interventions, and medication protocols have been used as interventions to prevent aggression and de-escalation. Although risk assessment and multimodal interventions have been shown to be effective in preventing aggressive behavior, no studies regarding their effectiveness in preventing escalation have been available.
18	Violence committed against nursing staff by patients in psychiatric outpatient settings	Jenni Konttila, Hanna-Mari Pesonen RN, Helvi Kyngäs. (2018) [24]	To elucidate, based on previous studies, the violence committed against nursing staff by patients in outpatient settings in adult psychiatry.	A literature search of the following databases was conducted: CINAHL (EBSCO), Ovid MEDLINE, and PsycARTICLES (Ovid). Search terms used were (community mental health OR community care), (outpatient setting*), (mental health setting* OR mental health service*), (nurs* OR service*), (psychiatric nurs* OR mental health nurs*), (mental health* OR psychiatric*), (violen* OR aggress*), (adult*), (outpatient). A total of 14 studies were included.	I	Staff members can become victims of violence for several reasons, including working conditions and physical and psychological factors related to the work environment (e.g., the characteristics of the physical environment), the number of staff supervisors, and the atmosphere of the workplace. The form of violence that occurred most frequently was verbal violence, with nurses at risk of violence in various situations. The incidence of violence has physical and psychological effects on the staff.
19	Behavioural, psychiatric and psychosocial factors associated with aggressive behaviour in adults with intellectual disabilities: A systematic review and narrative analysis	Natalie van den Akker, Marieke Kroezen, Jannelien Wieland, Annelieke Pasma, Ria Wolkorte. (2021) [27]	To provide an overview of the association between behavioural, psychiatric, and psychosocial factors and aggressive behaviour in adults with intellectual disability.	Embase, Medline, Web of Science, PsycINFO, Cochrane Central, CINAHL and Google Scholar were searched. A wide variety of the following search terms was used: intellectual disability, challenging behaviour and different terms for behavioural, psychiatric, and psychosocial factors. Studies on adults with intellectual disabilities are included. Those that focused only on biological factors and on the relationship between age, gender, and degree of intellectual disability and challenging behavior were excluded. In total, 38 studies were included.	M	Forms of aggressive behavior, such as physical, verbal, and self-injurious behavior, were interrelated. Although several studies have focused on the relationship between aggressive behavior and psychiatric disorders and symptoms, such a relationship has yet to be been fully elucidated. Psychosocial factors may have complex and conflicting effects on aggression.
* Letters in the scope items indicate the following; I: inpatient; C: community; M: medical field (no distinction between outpatient and inpatient).						
**〇 Summary of Studies on Rating Scales for Factors Associated with Harmful Behavior Toward Others**						
**No.**	**Title**	**Author (year)**	**Aim**	**Subjects**	**Scope** *****	**Summary of results and discussion**
20	Structured assessment of violence risk in schizophrenia and other psychiatric disorders: a systematic review of the validity, reliability, and item content of 10 available instruments	Jay P. Singh, Mark Serper, Jonathan Reinharth, and Seena Fazel. (2011) [31]	To undertake a systematic review on structured violence risk assessment tools in individuals with schizophrenia.	Risk assessment tools designed to predict the likelihood of community violence in psychiatric populations were identified using PsycINFO; Embase; Medline; the US National Criminal Justice Reference Service Abstracts; and combinations of the following Boolean keywords: violen*, risk, assess*, predict*, tool*, instrument*, measure*, mental*, and psychiatr*. After applying the inclusion and exclusion criteria, 10 of the 158 violence risk assessment tools identified were included.	I	Ten violence risk assessment tools for psychiatric disorders were analyzed. The validity and reliability of the tools were scored using a checklist, with HCR-20, HKT-30, VRAG, and VRS having satisfied the most criteria.
21	Violence risk assessment in psychiatric patients in China: A systematic review	Jiansong Zhou, Katrina Witt, Yutao Xiang, Xiaomin Zhu, Xiaoping Wang and Seena Fazel. (2016) [32]	To examine three main areas: (1) the current state of risk assessment research in China, (2) the instrument most frequently used to assess aggression and violence risk in China, and finally (3) whether these instruments are associated with a similar degree of predictive validity as found in Western samples.	Eight databases, including Medline, EMBASE, and PsycINFO, were searched using the following Kkey words: aggression, violence, psychopathology, and risk assessment/prediction. Studies conducted in mainland China that examined the reliability and validity of psychometric tools/risk assessment instruments designed to predict the likelihood of aggression or violence were included. Data with no reliability or validity were excluded. Overall, 30 studies were included.	I	None of the sudies reported information on the reliability or validity of the aggression tools. The risk assessment tools used in Western countries and China were analyzed. Five tools (BVC, PCL-R, HCR-20, V-RISK-10, and LSI-R) showed excellent reliability, whereas two tools (VRS and HCR-20) showed excellent reliability.
22	Workplace Violence Toward Mental Healthcare Workers Employed in Psychiatric Wards	Gabriele d’Ettorre, Vincenza Pellicani. (2017) [29]	To evaluate which topics have been focused on in the literature and which newly approach the concern of patient violence against health care workers employed in psychiatric inpatient wards,	Articles published in PubMed and Web of Science databases were retrieved using the following key words: violence, inpatient psychiatric units, mental health workers, assault, prediction, prevalence, occupational risk, safety measures, risk assessment, and risk management. A total of 64 studies were included.	I	The VSC, HCR-20, PCL-SV, DASA, V-Risk-10, and BPRS were used for risk assessment. Risk management included building staff–patient relationships and improving staff communication skills. Between 24% and 80% of the staff had been victims of violence, which caused physical and psychological effects.
23	Violence risk-assessment screening tools for acute care mental health settings	Kendra K Anderson, Carole E Jenson. (2019) [1]	To identify violence risk assessment screening tools that could be used in acute care mental health settings.	Violence risk assessment screening tools were identifed by narrowing the search to studies published within the past 5 years and to tools appropriate for acute care mental health settings. Eight violence risk-assessment screening tools were included, among which four were selected for validity and validity.	I	Four of the eight violence risk assessment tools were validated (START, BVC, DASA-IV, and V-RISK-10). The BVC and V-RISK-10 provided sufficient statistical information to consider their use in acute mental health settings and strongly support their use in mental health practice.
24	Violence risk assessment instruments in forensic psychiatric populations: a systematic review and meta-analysis	Maya G T Ogonah, Aida Seyedsalehi, Daniel Whiting, Seena Fazel. (2023) [30]	To conduct a systematic review and meta-analysis on the performance of risk assessment instruments used to predict interpersonal violence and crime in forensic psychiatric patient samples after discharge.	A systematic search was conducted to identify studies that measured the performance of risk assessment instruments in predicting the outcome of interpersonal violence and crime in forensic psychiatric samples post-discharge. Four databases (PsycINFO, Embase, Medline, and Global Health) were searched. Sex offenders referred for civil commitment were excluded. A total of 50 studies were included.	I	The predictive performance of 36 tools was evaluated; however, evidence on predictive performance of current violent risk assessment tools in forensic mental health has been heterogeneous. Regarding the predictive performance of the most common risk assessment tools, the H10, HCR-20 version 2, VRAG, and Static-99 showed similar performance in predicting violent recidivism.
* Letters in the scope items indicate the following; I: inpatient; C: community; M: medical field (no distinction between outpatient and inpatient).						

The asterisk (*) used in English search terms typically functions as a wildcard. It allows you to replace any string of characters and is commonly used in search engines or databases.

**Table 2 healthcare-13-00152-t002:** Factors related to the harmful behavior.

Category	Subcategory	Code	Risk Factor	Protective Factor	Literature (Table 1)
Background factors (demographic, personal history)	Demographic and environmental factors	Age	young (≦age 35)		3, 4, 10, 11, 18, 20, 21, 22
		Gender	Male		3, 4, 10, 11, 13, 14, 18, 20
		Global violence risk (social conditions and race)	Yes		13, 20
		Violence-prone environment	Yes		5
	History of self-injurious/other harmful behavior	History of violent	Yes		7, 10, 11, 13, 14, 18, 20, 22
		Suicidal ideation, threats, or previous suicide attempt	Yes		10, 13, 20
	Apprehension/criminal history	History of criminal/criminal activity	Yes		13, 20
		History of non-violent offending	Yes		20
		Criminal peers	Yes		20
	Violence victimization/experienced abuse	Experiences of violence victimization	Yes		7, 13, 18
		Experienced child/physical/sexual abuse	Yes		13, 20, 21
		Witnessing Violence	Yes		6
	Family history (mental illness/substance use problems/criminal involvement)	Family history of mental illness	Yes		20
		Family history of substance use problems	Yes		13, 20
		Parental history of criminal involvement	Yes		13
	Intelligence/history of school	Intelligence	Below average	Above average	3, 20
		History of school/school achievement	Low—Poor		10, 20
	Secure attachment and conduct problems during childhood	Lived with both biological parents during childhood	No		20
		Secure attachment in childhood/stable nurturing environment	Exist	Does not exist	1, 4, 6, 20, 21
		Conduct problems during childhood	Yes		6
Background factors (clinical aspects)	Mental illness/organic disorder (schizophrenia/personality disorder/mood disorder)	History of mental illness	Yes		3
		Existing mental illness	Yes		4, 6, 7, 10, 18, 19, 20
		Schizophrenia/schizoaffective disorder	Yes		9, 10, 14, 15, 18, 20, 22
		Personality disorder	Yes		6, 13, 18, 20
		Mood instability (mood disorder)	Yes		3, 20
		Anxiety disorder	Yes		20
		Intellectual disabilities	Yes		16, 19
		Head trauma/physical illness/neurological disorders	Yes		3, 4, 5, 8, 20
		History of treatment in hospital or correctional institution	Yes		11, 20
		Period of hospitalization	Long		7, 10
	Substance use problems (alcohol/drugs)	Substance use problems	Yes	No	7, 9
		Previous and/or current drug abuse	Yes	No	3, 4, 6, 10, 11, 13, 18, 20, 21, 22
		Previous and/or current alcohol abuse	Yes	No	4, 7, 10, 13, 14, 22
		Availability of alcohol/drugs	Yes		5
	Treatment status (treatment motivation/compliance/sensitivity)	Compliance/treatment motivation	Poor/Lack	Presence	2, 10, 13
		Steady progress of treatment/treatability	No		20, 21
		Pharmacological treatment motivation	No	Yes	18, 20
		Treatment in correctional and psychiatric facilities	Yes		3, 20
		Mandatory treatment or hospitalization	Yes		7, 10, 11, 14, 20
		Problems with the hospital where the patient is being treated	Yes		10
	Positive symptoms (e.g., hallucinations and delusions)	Hallucinations/delusions	Yes		18
		Psychosomatic symptoms	Yes		3, 10, 20, 21
Situational factors (daily life status)	Work/economic situation	Work/work training	No	Yes	3, 10, 20
		Economic situation (income)/financial management	Instability	Stability	20
		Poverty	Yes		5, 10
	Basic self-care independence	Self-care/ability to perform daily chores	Lack		20
		Socioeconomic status	Lower		13
		Residence	No		3, 5, 13, 20
Situational factors (social connection status)	Presence and interaction with family and friends	Cohabitee	No	Yes	20
		Marital/partner	Unmarried, divorced	Yes	10, 11, 20
		Family/friends	No	Yes	1, 5, 20
	Social connections and access to public and private support systems	Acceptance of social support and assistance	No		4, 18, 20
		Social support/network	Lack	Yes	4, 21
		Social networking	Cannot		20
		External control	No	Yes	20
		Social and relational skills	No	Yes	1, 20
		Social connections/neighborhoods	No		5, 20
		Building stable and close relationships	Cannot	Can	2, 20
		Low levels of social cohesion	Yes		5
Psychological factors (thinking, cognition, and personality)	Self-awareness/foresight	Understanding of mental illness and treatment needs/insight into illness and/or behavior	Lack	Yes	13, 20
		Ability to develop a feasibility plan	Lack	Yes	20
		Awareness of own behavior, thoughts, health status	No		20
		Failure on conditional release	Yes		20
	Ability to put yourself in other people’s shoes (cooperation/empathy)	Empathy/cooperativeness	Lack	Presence	1, 20
		Remorse/guilt feelings	Lack		21
		Interpersonal and self aggression/aggressive behavior	Yes		1, 3, 11, 13, 20, 21
		Suspiciousness/hostility/hostile reaction style	Yes		20, 22
		Negative attitudes	Yes		20
		Irresponsible. Does not accept responsibility	Yes		20
		Agreement on conditions/rules	No	Yes	2, 20
		Acculturation issues	Yes		20
		Self-esteem, narcissism, and grandiosity	Strong		10
	Impulsivity/emotional instability	Impulsivity/poor behavioural controls	Yes	No	1, 13, 21
		Presence of instability/exposure to instability	Yes		20
		Impulse control/behavior control	Cannot	Can	1, 20
		Perceptual sensitivity	Yes		21
	Vulnerability to stress/coping skills	Leisure/recreational activities	No	Yes	20
		Feelings of frustration	Yes		4
	Abnormalities or distortions in personality, thinking, and behavior	Homicidal thoughts	Yes		20
		Psychopathy	Yes		13
		Sexual deviance and sexual escapism	Yes		20
		Cognitive distortion/distortion of thought	Yes		1
		Distortion of problem-solving methods	Yes		1
Antecedents of harmful behavior	Imminent risk of harm to others and threatening behavior/attitude	Overall violence risk	Yes		20
		Irritable	Yes		3
		Boisterous	Yes		3
		Agitated behavior	Yes		3, 13, 17
		Confused	Yes		10, 20
		Verbal threats and physical threats	Yes		3
		Ability to comply with directions and/or reasonable limit setting	Yes		3
		Loud talking	Yes		3
		Use of weapons (harmful) and attacking objects	Yes		3, 20
Triggers of harmful behavior	Triggers of the harmful behavior	Stressing circumstances	Yes		3, 20
		Loss-like life events (divorce, bereavement, unemployment)	Yes		3, 21
		Requests are denied/required	Yes		3
		Negative/negative attitude	Yes		18
		Victims/targets	Exist		5, 18, 20

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
