# Peer review of "Situational Factors Impacting Harmful Behavior Towards Others Related to Mental Health in the Community and Their Associations: A Scoping Review Based on Systematic Reviews"

_healthcare, 2025, doi:10.3390/healthcare13020152_

Round 1

Reviewer 1 Report

Comments and Suggestions for Authors

The current paper (Factors Leading to Harmful Behavior Toward Others and Their Associations: A Scoping Review) is a scoping review aimed to identify factors associated with harmful behavior toward others based on existing research.

Although it is undoubtedly a very important study, there is one major and key concern related to its design. This major concern relates to the research question that underlies the study. It is not clear from the introduction to the method (and consequently to results and discussion) what is the studies’ population to be selected for the scoping review. This hampers the quality of the massive work authors have made, as there are several studies included that although focused on harmful behaviors are to disperse (in terms of population; e.g., forensic, inpatient) for comparisons and for build a coherent rationale. In turn, even if the goal was to include a broader range of studies/populations, I am afraid that the research equation could be puzzled to gather and/or difficult to target all the literature on the topic.

Unfortunately, this critical point hampers the quality of this work, which, again, may posit a very interesting topic if the research question that based the searches was sufficiently clear.

Hope you find this review useful and constructive to strengthen your research in the future

Best wishes

Author Response

Thank you very much for taking the time to review this manuscript. Please see the attachment.  

Reviewer 2 Report

Comments and Suggestions for Authors

Thank you for your submission, this is an important topic and I look forward to reading the completed version.  The title is ‘Factors Leading to Harmful Behavior Toward Others and Their 2 Associations: A Scoping Review.’ However, your search terms are violence, aggression, problem behavior, and workplace violence only.

 Perhaps it may be beneficial to label your title to ‘Situational factors that impact harmful behaviours towards others’ Or ‘Risk Factors leading to…’. I feel this title needs to be more specific. I  further note that you have listed several interventions to manage violence in the workplace such as risk assessment tools, although more focus on the factors that impact a person to commit a harmful behaviour would be beneficial?

For example, in the your introduction you have identified several factors ie situational and background factors that impact an individual to cause violence, then focus on the interventions. Table 4 has the most merit in your work as it is not necessary to list definitions of violence.

I note that the majority of papers that were sought included violence management and a notable study with L. Bowers has already acknowledged factors surrounding violence management over the past 20 years ie Safewards.

Line 45-56- There is no referencing in-text, please ensure you acknowledge your paragraph on MH welfare specialists, harmful behaviour and noted hospital/ community environments.

Overall, the subject being explored is not unknown and extensive work has already been carried out. This paper needs aligning to the actual factors highlighted in table 4. The focus of the paper needs work starting at the title, introduction, body and discussion. If you want to explore violence management interventions then this could be a potential second paper?

Author Response

(The authors gave the same response as above.)

Reviewer 3 Report

Comments and Suggestions for Authors

Thanks to the authors for sharing their manuscript. I think the idea of the study is good, the results are presented fully and meaningfully. I have a few doubts about the design of the study:

Firstly, it is necessary to clearly define the inclusion and exclusion criteria, as well as the terms included in the analysis. Why do the authors single out workplace violence, but not other types of violence (physical, psychological, sexual)? Why did the authors exclude child and elderly violence? This information is shown in the figure, but it would be good to put it in the text.

Secondly, I am concerned that the authors have accessed only two databases (PubMed and Scopus). Scoping reviews are usually based on a larger number of databases. Perhaps the authors have a justification for their choice, but at this point it is necessary to write in the manuscript.

A small comment: please correct the error in the figure (lines 168-174: it should be Scopus instead of Scopas).

If the authors add inclusion and exclusion criteria to the manuscript, as well as justify the choice of databases, I recommend the manuscript for publication.

Sincerely yours,

the reviewer.

Author Response

(The authors gave the same response as above.)

Round 2

Reviewer 1 Report

Comments and Suggestions for Authors

Authors did a good job in addressing the issues and in focusing and clarifying previous queries.

Author Response

Thank you very much for taking the time to review this manuscript. 

Reviewer 2 Report

Comments and Suggestions for Authors

This is much improved and you have acknowledged and addressed my comments in report 1. Well done. Accept manuscript in current form

Author Response

(The authors gave the same response as above.)
